# The experience of living with patellofemoral pain—loss, confusion and fear-avoidance: a UK qualitative study

Benjamin E Smith,[1,2] Fiona Moffatt,[3] Paul Hendrick,[3] Marcus Bateman,[1] Michael Skovdal Rathleff,[4,5] James Selfe,[6] Toby O Smith,[7] Pip Logan[2]

For numbered affiliations see end of article.

**Correspondence to**
Benjamin E Smith;
benjamin.smith3@nhs.net

## ABSTRACT

**Objectives** To investigate the experience of living with patellofemoral pain (PFP).

**Design** Qualitative study design using semistructured interviews and analysed thematically using the guidelines set out by Braun and Clarke.

**Setting** A National Health Service physiotherapy clinic within a large UK teaching hospital.

**Participants** A convenience sample of 10 participants, aged between 18 and 40 years, with a diagnosis of PFP and on a physiotherapy waiting list, prior to starting physiotherapy.

**Results** Participants offered rich and detailed accounts of the impact and lived experience of PFP, including loss of physical and functional ability; loss of self-identity; pain-related confusion and difficulty making sense of their pain; pain-related fear, including fear-avoidance and 'damage' beliefs; inappropriate coping strategies and fear of the future. The five major themes that emerged from the data were: (1) impact on self; (2) uncertainty, confusion and sense making; (3) exercise and activity beliefs; (4) behavioural coping strategies and (5) expectations of the future.

**Conclusions** These findings offer an insight into the lived experience of individuals with PFP. Previous literature has focused on pain and biomechanics, rather than the individual experience, attached meanings and any wider context within a sociocultural perspective. Our findings suggest that future research is warranted into biopsychosocial targeted interventions aimed at the beliefs and pain-related fear for people with PFP. The current consensus that best-evidence treatments consisting of hip and knee strengthening may not be adequate to address the fears and beliefs identified in the current study. Further qualitative research may be warranted on the impact and interpretation of medical terminology commonly used with this patient group, for example, 'weakness' and 'patellar mal-tracking' and its impact and interpretation by patients.

**Trial registration number** ISRCTN35272486; Pre-results.

## INTRODUCTION

Patellofemoral pain (PFP) is one of the most common and costly forms of knee pain.[1–3] It has an estimated prevalence of 23% in the general population in the UK.[1] Symptoms typically include retropatellar or diffuse peripatellar pain, aggravated by activities that load the joint, such as climbing and descending stairs, squatting and running.[4]

Historically, PFP has been labelled a 'benign, self-limiting condition', that improves over time with little intervention indicated.[5] However, this belief has recently been challenged with data suggesting that the overall long-term prognosis for the majority of patients with PFP is poor.[6] Only one-third of patients are pain-free 1 year after diagnosis,[6] and 91% still report pain and dysfunction 4 years postdiagnosis.[7] Quantitative data suggest that some patients withdraw from participation in physical activities[8 9] and may develop associated psychological distress, such as fear-avoidance and catastrophising thoughts in relation to their knee pain.[10–12]

The biopsychosocial model of persistent pain has recognised that psychological factors such as fear and catastrophising can, through changes to behaviour, modulate physiological responses to pain with the development and maintenance of persistent pain.[13–17] Psychological distress has been identified in low back pain and tendon pain populations through systematic reviews[18 19] and qualitative methods

### Strengths and limitations of this study

► This is the first study to use a qualitative method of inquiry on the experience of people living with patellofemoral pain.
► Two authors independently coded all transcripts, and a clear, transparent and reproducible methodological approach was used in the thematic analysis.
► For pragmatic reasons, a convenience sampling technique was used.

in low back and shoulder populations[20–22]; however, to our knowledge, this has not been investigated in PFP. Advocates of qualitative research methods suggest that qualitative inquiry can disclose the experience of people with pain and, therefore, be used to understand patient motivation, social engagement and provide a wealth of information about the sociocultural context to pain.[23 24] Contemporary models of persistent pain have identified the importance of thinking beyond muscles and joints,[25] and qualitative inquiry can provide an insight that may lead to development of ideas and hypothesis generation within the context of the biopsychosocial model of pain. No study using qualitative methods has been published regarding PFP. Therefore, the aim of this study was to give a more detailed account of the experience of people living with PFP, seeking secondary care within the UK.

## METHOD

In order to address gaps in the literature, this research focused on identifying themes within the participants' experience of living with PFP. A qualitative interpretive description design was chosen as an appropriate methodological approach.[26] Thematic analysis is the most appropriate method for this type of inquiry, as codes and themes can be created inductively to capture meaning and content without prior preconceptions allowing flexibility to generate a rich and detailed account of the data.[27]

In this study, data were analysed thematically using the guidelines set out by Braun and Clarke[27] and were reported in line with the COnsolidated criteria for REporting Qualitative research checklist .[28]

Braun and Clarke[27] describe a multistage approach to thematic data analysis; demonstrating clear distinction of the thematic approach, while allowing for the inherent flexibility in the process. They reasoned that a thematic analysis can be conducted from both realist and constructionist paradigms, although with differing outcomes. A realist approach allows theories about individual motivation and meaning to be developed, since the epistemological position is that there is a unidirectional relationship between meaning, experience and language.[27] A constructionist perspective differs, as meaning and experience are socially produced and knowledge a human and social construct; therefore, theories about individual motivation and meaning are inappropriate, and theories focus instead on sociocultural contexts.[27] This study did not set out to prove or disprove a hypotheses; it set out to generate new data from which an understanding of living with PFP might be developed. The authors wanted to take an epistemological position that recognises the experience at an individual level, and any meanings attached, while considering the wider context within a sociocultural perspective. Sitting central on the spectrum of realism and constructivism, this position is described as 'contextualist' by Braun and Clarke.[27]

## RECRUITMENT

A convenience sample of 10 participants with a diagnosis of PFP was recruited from the National Health Service physiotherapy waiting list. Based on similar studies of other musculoskeletal conditions, we anticipated that this sample size would be sufficient to reach data saturation and was agreed a priori.[22 29] Participants were initially contacted by mail and followed up by a telephone call (BES). Thirty-four information sheets were sent out, and 24 potential participants were contacted by telephone; two could not make the interview before physiotherapy was due to start; five people physiotherapy had already commenced; one reported resolution of symptoms and six declined to participate. Inclusion criteria were participants aged 18 to 40 years with signs and symptoms of PFP, defined as anterior or retropatellar pain reported on at least two of the following activities: prolonged sitting, ascending or descending stairs, squatting, jumping and running.[4] These were prescreened during an initial telephone conversation. Exclusion criteria included: previous knee surgery, awaiting lower limb surgery, knee ligamentous instability, history of patellar dislocation, true knee locking or giving way, reasons to suspect systemic pathology or acute illness, pregnancy or breast feeding, patellar or iliotibial tract tendinopathy and those not able to speak or understand English. The exclusion criteria were screened prior to consent being taken (BES).

## DATA COLLECTION

Participants were offered interviews at their home or in a hospital-based physiotherapy department; all opted to be interviewed at the hospital. On arrival, the researcher (BES) introduced himself as a physiotherapist working in that department and also a researcher conducting a PhD. The researcher explained the aims of the study. Written consent and verbal consent were taken to start recording.

With reference to previous literature on low back pain, shoulder pain and tendon pain,[20–22] semistructured interviews were designed by the researchers using a topic guideline with prompts to explore participants' experience of: living with PFP; past healthcare management; their interpretation of causation of their pain; beliefs, attitudes and behaviour in relation to their pain and expectations for the future. The semistructured interviews allowed for a flexible interview, in a two-way conversation, allowing new ideas to be developed as they were brought up.

The researcher also maintained a reflective journal, noting down initial thoughts and ideas after each interview.[22] This identified that early interviews raised issues

about other (past and present) musculoskeletal pain and specific coping strategies employed by participants for their PFP. These were therefore incorporated into subsequent interview schedules.

## DATA ANALYSIS

All audio files were collected and transcribed verbatim (BES). During transcription, initial thoughts and ideas were noted in the reflective journal. Audio files were listened to several times to check for accuracy, and transcriptions were read and re-read a number of times; this initial process of data familiarisation allowed for 'data immersion' by the researchers and generation of preliminary ideas.[27] Data coding then identified and coded pertinent features of the data giving equal priority over the whole dataset. These steps were independently conducted by two researchers (BES and FM) who met to compare codes and develop agreement on the grouping of codes into themes. The generated themes were reviewed and refined, ensuring that they explained the data in relation to the coded data and the whole dataset. The researchers then consulted on the final two stages; themes and subthemes were named and defined to demonstrate a clear narrative, using compelling extracts as illustrations. Consideration was given to each theme individually, but also to how they related to the dataset as a whole and other themes (see online supplementary file 1 for the code book).[27]

Data were organised and analysed using QSR International's NVivo V.11. After 10 interviews, it was determined by the researchers that data saturation had occurred as no new thoughts or concepts were generated in the later interviews.

## RESULTS

Participants ranged from 26 to 37 years of age (mean age 30.6 years), with a diagnosis of PFP for a mean duration of 78 months (range: 3 months to 16 years). For participants'

characteristics, see table 1. The interviews ranged from 13 to 43 min (mean time: 27 min).

The first theme that emerged from the data, impact on self, describes the participants' sense of loss, in relation to their self and self-identity. The further themes that emerged describe how the participants deal with this loss in a climate of uncertainty, how they understand or make decisions regarding exercise/activity and pain management and how they prognosticate for the future. Data are presented to demonstrate the range and meaning to each theme.

### Theme 1: impact on self

Participants offered rich and detailed accounts of the impact and lived experience of PFP. Loss of self and loss of self-identity were evident in the stories told by many of the participants in this study. Self and self-identity are different concepts about ways in which individuals evaluate and interpret themselves; they are nested elements that are shaped by the contexts of individuals' lives, with direct influence on decisions and behaviours.[30] Self, in its broader sense, can be defined as one's individuality and process of making sense of the world around them; it is a cognitive structure that defines one's sense of worth.[31] Self-identity, however, is the cognitive structure of internalised meanings and expectations associated with one's position and role within a social network.[32]

Symptoms affected all participants' daily life, with pain being a pervasive and disruptive feature of their day, with resulting loss of physical ability:

> I struggle at work, bending down to get bottom shelf and getting back up, I literally have to hold onto the table to pull myself up. I can't do it off just my knees. [P7].

> Yeah, well, it's a pain really because I'm walking around. I'm very stiff with that leg. Going up the stairs, down the stairs at work, getting out of a chair, getting into the car. [P6].

| Table 1 | Characteristics of participants | | | |
|---|---|---|---|---|
| Participant no | Gender | Age | Duration of symptoms (m) | Type of employment |
| 1 | F | 26 | 60 | Healthcare worker |
| 2 | M | 33 | 60 | Builder |
| 3 | M | 37 | 8 | Office worker |
| 4 | F | 26 | 192 | Healthcare worker |
| 5 | F | 34 | 36 | Office worker |
| 6 | F | 27 | 84 | Waitress |
| 7 | F | 28 | 120 | Technician |
| 8 | M | 29 | 36 | Office worker |
| 9 | F | 36 | 3 | Office worker |
| 10 | F | 30 | 180 | Office worker |

F, female; M, male; m, months.

Several participants described the negative impact of PFP on their mental well-being, with subsequent loss of self-identity:

> I would say the reason I got my horse was because I have mental health problems and so having a horse is my routine, structure, thing that I look forward to doing. The positive in my life. And having the knee problem makes that, makes that, not so effective. You can't do, what I imagined I would be able to do. [P4].

Physical activity has been identified as a key quality of life domain, and the one most affected among patients with persistent pain.[33] Loss of activities for these participants included: walking, exercise, driving, holidays, time with family and friends, playing with children, duties at work and kneeling. These loss of activities directly affected participants' role and position within their social network, triggering feelings of loss of self-identity. For example, a number of participants explained how PFP affected their work and made them question their career aspirations:

> I would say, it makes me like wonder, if I can do the job, not at this point but maybe when I get older and older, maybe I won't be able to do it. [P4].

Judgemental attitudes from colleagues, friends or family were described by a number of participants, with subsequent feelings of loss of self-identity, acting as moderators to low moods and feelings of premature ageing:

> They're saying that I'm a grandma. They say, 'Yeah. If you were a horse, they'd put you down (laughter). Just joking me, but obviously, it has affected me in the way that I've had to go out of work to go over to get physio. And I have had this time off, so I don't know if they're a bit, 'Well, it's not that bad.' Because day-to-day I try to be as normal as I can. [P9].

Loss of significant relationships has emerged as a key aspect of loss in previous studies of patients with persistent pain[34–36]; and disruption to important and meaningful relationships was a strong and common theme found in patients with PFP. For example:

> I've missed out of things over the years, spending time with friends, spending time with family and that kind of thing, because I've not been able to do it. [P6].

As identified by the above extracts, PFP had a compelling and far reaching impact on the participants and their lives. The pain and its disruption to life, loss of self-identify and loss of relationships were themes that emerged from the data.

### Theme 2: uncertainty, confusion and sense making

Confusion and sense making formed a central part in the lives of the participants, with a strong desire from all to elucidate the cause of their pain.

> If I could find out what it was that was causing the pain, then you hope it would be gone within a year.

But because we don't really know what's caused it, it's kinda trial and error. So I don't really know. [P1].

The predominant focus of the participants' beliefs and attempts at making sense of their pain was that biomechanical factors were causative, with individuals trying to link these factors to the development and maintenance of their pain.

> My running technique or, I'm not sure. I'm not sure about that. I'm not sure. I think that's one thing, maybe something to do with the running technique, or something, or something to do with that. [P8].

Furthermore, confusion was also related to the episodic nature of the symptoms, with participants attempting to relate 'flare-ups' to the same biomedical factors.

A number of participants told stories of structural and biomedical beliefs becoming deep-rooted and established when reinforced. For example, one participant recounted multiple encounters with healthcare practitioners who influenced and reinforced her structural belief.

> The work physio guy said to me that he thinks that my heels have maybe gone in which has then pulled my kneecap out of alignment. So instead of going smoothly over the joint where it's supposed to, that it's probably moving over the bone and that's the sharp pain that I'm feeling. Which did make sense because it, like I said, felt like I'd got a rock underneath my kneecap at some stage. [P9].

Some participants remembered biomechanical focused diagnoses they had been given by a healthcare practitioner they had seen many years in the past; highlighting the power and lasting influence healthcare practitioners have on their patients. For example, one participant remembered the diagnosis she had received from a healthcare practitioner over 10 years ago:

> I had to go to the hospital once to have x-rays… I don't know if he [doctor] was trying to scare me into doing some exercise or something, but he basically said the only thing they could do is break both of my thighs and twist them a bit and then heal them back together. And it would take me years to get back to walking properly. [P4].

Joint noises are a common feature of normal joint movement;[37] however, participants commonly reported distress and confusion at joint noises, often finding healthcare practitioners' explanations inadequate.

> It was the noise that was concerning me more than the pain. I'm used to hurting. I'm too small to play rugby for a start, and I'd been fighting for 20 years, so, erm, it's one of those, you get used to the pain, but it's just the noise. When you start, you sort of [say] no, that's not right. [P3].

This was in agreement with previous research, which identified negative emotions and inaccurate aetiological beliefs with joint noises in patients with PFP.[37]

Expressly linked to participants' confusion and need to find the cause of their pain was also a strong desire to pursue radiological imaging and feelings of not being taken fully seriously by the healthcare profession when this was not forthcoming.

> I want to know exactly what the problem is. Obviously, the doctor said, previously going back, they said tendonitis, and now they're saying it's runner's knee or whatever. But you know, it's still like, is that 100%, are you sure that's what it is? Because I was going to ask the doctor to send me for a MRI… [P8].

Previous research has linked poor outcomes with radiological imaging in populations with low back pain, suggesting that an overuse of imaging has a detrimental effect on outcomes.[38] There was one example of the resulting radiological findings compounding the confusion and distrust, for example, participant six explained her feelings on a normal MRI finding as:

> I mean I was a bit concerned, because they didn't turn around and say, you have hurt it, but it's not major but this is what you've done, but they didn't actually, they said nothing's wrong, take the knee brace off, and carry on. [I was] almost deflated, because I was like wanting to know why it was hurting, but they weren't explaining any of that to me. So it's a bit like, difficult. [P6].

Another participant's story demonstrates the negative impact of discordance between healthcare practitioners' diagnosis and advice, further compounding confusion and mistrust:

> Well, it makes you wonder then which one to believe, because I'm like,' Well okay, he's told me not to do anything until I'm pain-free, because he doesn't want me to aggravate it,' but when, when I came here, and obviously they said that it would probably be best to start putting an impact on it again… [P9].

The sense-making processes that participants described were established from past experience of healthcare treatment, past experience of pain and cultural beliefs around structure and pain.

### Theme 3: exercise and activity beliefs

All participants identified specific beliefs regarding barriers to exercise and activity. These were informed by factors relating to: diagnosis uncertainty, cultural beliefs around pain, fear-avoidant behaviours and the iatrogenic effect of healthcare.

Diagnosis uncertainty contributed to participants' beliefs regarding exercise and activity. In particular, it underpinned a dilemma regarding the relationship between activity and potential harm:

> It's 'are you making it worse?' And that's the crux of it really. As I'm doing it and thinking, 'if this is hurting, should I really be doing this, or shall I pack this in and do something else?' But it's the not knowing… [P5].

Cultural beliefs around pain being a direct sign of tissue damage were evident in a large proportion of the participants' narratives, resulting in negative behaviour towards exercise and activity.

> … with me it's always been, if something hurt it because your body's telling you if you do that you're going to cause more injury. You'll make things worse. [P6].

Associated with the cultural beliefs on pain and damage was the resultant fear-avoidant behaviour. Participants frequently contradicted themselves; however, many participants would express the sentiment that they would not let the pain stop them from doing what they wanted to do, yet demonstrated clear activity withdrawal.

> So for example, we went to [holiday resort] last year; on your feet all day, walking miles and miles, I would be, like, in tears by the end of the day. I wouldn't let it stop me the next day because I would be, like, I'm doing this [P4].

> When I was in [holiday resort]; a couple of days I didn't go out and I stayed back at the hotel. Because I couldn't do it, I needed to rest. [P4].

A predominant subtheme was the association of sports and exercise, even in the absence of pain, as a potential precursor to future joint pain and 'damage'. Some participants attributed their current PFP to past sporting activities, despite no obvious mechanisms of injury.

> Yeah. Obviously it stems from doing long distance running. [P7].

A number of participants discussed the direct impact of healthcare practitioner's advice and diagnosis labelling on their exercise and activity levels, suggesting an iatrogenic effect of healthcare for PFP patients.

> I have been told by doctors before I shouldn't run because it would jar my knee and shouldn't run or walk on an uneven surface because it will wonk my knee from side to side. [P4].

> But then when I started the physio at work and he told me that I shouldn't walk or that I shouldn't swim because he just wanted to obviously manipulate it and get me pain-free before I did anything that could possibly aggravate it. So I stopped. [P9].

This theme identified a number of beliefs associated as a barrier to activity and exercise engagement. These included diagnosis uncertainty, cultural beliefs around pain, fear-avoidant behaviours and the iatrogenic effect of healthcare.

## Theme 4: behavioural coping strategies

A central coping strategy for participants of this study was the concept of rest. Many of them associated rest, and avoidance of activity, with the idea that time was necessary for the healing process, and that aggravating activities should be avoided.

> I try, obviously, sit down as much as I can. [P4].

One participant expressed an expectation that healthcare professionals would advise him not to continue with activity and exercise:

> R: So you think physios would say no [to keep physically active]?
>
> P8: Physios would probably say no. Yeah, you shouldn't do it.

Another common coping strategy was postural adjustments; participants often talked of preferred sitting positions in relation to avoiding knee flexion.

In keeping with previous research on the high levels of analgesic use in patients with PFP,[7] a common narrative shared with participants was the use of analgesics, with some acknowledging they were not effective.

> I have had some strong painkillers from the doctors. They gave me some naproxen and some codeine to manage it when it was at its worst but I try not to take them. [P9].

The use of knee supports was also common in the self-management strategies employed by the participants.

> If it hurts, it hurts. I'll try and strap my knee up. Because if I know I'm going harder in like gym classes, I'll strap my knees up before I go. And then when I get too much pain, I'll stop the exercise. [P10].

## Theme 5: expectations of the future

A number of participants expressed views, which could be contextualised as an external locus of control, with expectations of passive physiotherapeutic treatment options.

> I would presume manipulation of muscles groups, joints and tendons. [P3].

Even though the majority of participants expressed negative views about the future, they all expressed a desire to be pain free, over and above any functional improvements.

> R: With the physio, what would you class as a success?
>
> P8: Getting rid of the pain.

Nine of the 10 participants held negative beliefs about the future, particularly in relation to prognostic prediction following their referral to physiotherapy.

> But then when I'm going up the stairs and it hurts it does concern me that it's going to be every day for the rest of my life I'm going to be struggling to walk upstairs. And then I think about getting old, and I think I'm going to end up with a stair lift and living downstairs and that sort of thing. [P1].

> [the pain is] definitely preying on my mind. Is it gonna stop me from going into the police, is that gonna stop me doing the things I want to do later on in life? So yeah, it does prey on my mind a little bit. [P6].

Central to their negative beliefs about the future and their prognosis was low self-efficacy. Participants felt that they had very little control over their symptoms.

> [In] my head, my thought process is I just hate it. Do an operation. Get rid of it. In my head, and obviously not being from the medical profession, but I'm just like, 'Just get rid of the pain however it can be done. [P8].

> Yes, I'm 37 now and they feel older than that. You just get that feeling, don't you, I've bounced back from lots of injuries before but this is the one that is making me think. You know, when this gets cold I can feel it, and thinking there's already arthritis there, I'm in trouble, it sets the brain going. [P3].

Low expectation of physiotherapy and past physiotherapy failed treatments were also a core theme within future expectations.

> R: Have you got any expectations of what might happen when you walk in to see the physio?
>
> P10: I expect them to turn around and say physio can't help.

> When I did get the physiotherapy it kinda didn't really do anything anyway. So it just made me think, it's pointless, 'cause they was trying to remove the fluid from out my knee, that like I say, made it worse to begin with. She did say your knees will feel sore, but it went back to how it was anyway, so, it just seemed like a pointless process. [P7].

There was one exception, with one participant having positive outlook to the future and their physiotherapy referral.

> Oh yeah, I think it will get better. Yeah, I'd go for the better option. [P9].

The main subthemes that emerged under the future were: beliefs that their pain will get worse, external locus of control with regard to treatment, low self-efficacy, poor opinion of physiotherapy and previous failed physiotherapy treatments and an overwhelming desire to be pain free, over and above any practical goals for rehabilitation.

## DISCUSSION
### Main findings

Quantitative research methodologies dominate the literature for PFP. This is the first study to use a qualitative method of inquiry to gain data on the experiences

of people living with PFP. The five major themes that emerged from the data were: (1) impact on self; (2) uncertainty, confusion and sense making; (3) exercise and activity beliefs; (4) behavioural coping strategies and (5) expectations of the future.

A key finding of this study is that loss of physical ability is profound and considerable and plays a significant role in participants' lives; despite previous research suggesting that PFP is a benign and self-limiting condition.[5] An inability to continue with significant and meaningful activities has been identified as a cause of anxiety in people with persistent pain.[39] Persistent pain interrupts behaviour and a person's self-identity by affecting a sense of who they are and what they might become.[40] As a result, lives are socially and environmentally restricted by persistent interruptions, or an inability to complete, or even attempt important tasks and activities.[40] With changes and loss of participants' position and role, for example, with employment or family duties, the internalised meanings and expectations associated with one's self-identity are further threatened.[32]

Participants expressed intense confusion around their pain and symptoms. For instance, the causative reasons were elusive and troubling, as too was the ability to predict and control the pain intensity; and any attempts that participants made at understanding were firmly within the biomechanical sphere of reasoning. An inability to make sense of pain and the process associated with sense-making and pain-related fear has been proposed in low back pain populations.[41] Previous research has identified that an inability to make sense of pain places 'lives on hold'[42] and may lead to more 'catastrophising'.[43]

There remains scientific debate and uncertainty around the underlying aetiology of PFP,[44] and there is a large variation in the way PFP is managed by physiotherapists in the UK.[45] The majority of participants in this study had previous experience of healthcare management for PFP suggesting that variation in healthcare treatment may have a negative impact on the patients' lived experience. Historically, the biomedical model of pain establishes a direct relationship between tissue structure and pain,[46] and participants characteristically attributed their pain to structure and/or anatomical problems. However, several studies have recently demonstrated that structural abnormalities of the patellofemoral joint on MRI are not associated with PFP.[47 48] Three participants had no previous healthcare management for PFP, but nevertheless gave a biomechanical/structural cause for their pain; all three had previous physiotherapy for other pain conditions, including back, hips and ankles. This may suggest that exposure to biomechanical approaches to the management of musculoskeletal pain in general could, potentially, have a carry-over to other locations of pain, with a negative effect.

The iatrogenic effect of healthcare is an emerging field of research in the low back pain population.[38 49] This study is the first to find such a theme in patients with PFP. These findings are consistent with recent research that showed that the majority of UK physiotherapists would advise their patients not to continue with exercises if they experienced any pain.[45] The fear-avoidance model of pain is well established with patients with persistent pain, particularly persistent low back pain[17]; additionally, research has shown that fear-avoidance behaviour may also exist with clinicians.[25 45 50] The central concept of the model is cognitions and emotions that underpin fear of the pain; fears about potential physical activities exacerbating the pain and further 'damaging' bodies. The fear leads to safety seeking behaviours and hypervigilance that paradoxically maintains or exacerbates the pain and disability.[22] In contrast, if pain is perceived in a non-threatening way, patients are likely to maintain physical activity levels, through which recovery can be achieved.[51 52] All of the 10 participants in this study described fear-avoidant behaviour at some stage of their interview. This is the first study, which we know of, that identifies this behaviour in patients with a diagnosis of PFP.

PFP is often described as an 'overuse' injury,[53] and these data seem to be consistent with the patients' belief and behaviour with a definition more aligned with the English language meaning of 'overuse'. Contemporary thinking in relation to injury risk challenges the idea that PFP is simply an 'overuse' injury, with evidence suggesting that persistent and long-term underuse may be a risk factor, with consistent exposure to tissue load being considered one method of management.[54] The fear-avoidant behaviours revealed within this study would therefore be seen as negative pain behaviour, with long-term detrimental consequences.

A key finding of this research is the low expectation for the future and low self-efficacy demonstrated by the majority of the participants that could be conceptualised as 'catastrophising'. Catastrophising is conceptually within the same model of pain behaviour as fear-avoidance, with large-scale overlap.[19] Low self-efficacy, fear of the future and catastrophising are common findings in patients with persistent pain.[24 55] The National Institute of Health and Care Excellence describes pain as a complex biopsychosocial issue, associated with expectations, self-efficacy, mood and coping abilities.[56] In addition, it has been shown that self-efficacy is a strong predictor of successful outcome, irrespective of the intervention delivered, for patients with persistent pain; suggesting that rehabilitation programmes for persistent musculoskeletal pain should be designed with the aim of improving self-efficacy.[57]

## Clinical and research implications

This study established that a sample of patients with PFP demonstrated: pain-related fear, such as fear-avoidance, damage beliefs, difficulty with making sense of their pain, low self-efficacy and fear of the future.

The current consensus that best evidence treatments consisting of hip and knee strengthening may not be adequate to address the fears and beliefs identified in the current study. Future studies are needed to explore

biopsychosocial targeted interventions for this population, particularly in relation to pain experienced by patients during exercise, followed by efficacy and effectiveness trials. Interventions may be patient education packages and self-management strategies targeting self-efficacy and physical activity. Furthermore, future qualitative work will be beneficial to understand the role of medical terminology commonly used with this patient group, for example, 'weakness' and 'patellar mal-tracking',[45] and its impact and interpretation by patients.

### Study limitations and strengths

Two authors independently coded all transcripts, and this study employed a clear, transparent and reproducible methodological approach to data analysis. The authors make it clear that their clinical and research experience lies within the biopsychosocial framework of musculoskeletal pain and this study forms part of a larger body of research looking at pain education, self-management strategies and exercise interventions for individuals with PFP.[58] It is worth noting that the interviewer made it explicit to the participants that he was a physiotherapist working in the department conducting the research; indeed, a number of them did proceed to ask clinical questions about their condition, highlighting a power dynamic between the interviewer and participant. Furthermore, it is important to note that recruitment took place in the same department that the researcher was working as a physiotherapist. This may, in part, have influenced participants' inclination to take part and also their responses.

The main limitation of this study is that for pragmatic reasons, a convenience sampling technique was used. It is possible that this sample may differ from other samples within the UK, and how representative these findings are to the greater population of individuals with PFP is unknown. A purposive sampling technique may have better represented sociodemographic groups or targeted identifiable subgroups.

### CONCLUSION

These findings offer an insight into the experience of individuals living with PFP. Previous literature has focused on pain and biomechanics, rather than the individual experience, attached meanings and any wider context within a sociocultural perspective. The participants provided rich and detailed narratives of loss of physical and functional ability; loss of self-identity; pain-related confusion and difficulty making sense of their pain; pain-related fear, including fear-avoidance and 'damage' beliefs; inappropriate coping strategies and fear of the future. Our findings suggest that future research is warranted into biopsychosocial targeted interventions and the impact and interpretation of medical terminology.

**Author affiliations**
¹Physiotherapy Department, Derby Teaching Hospitals NHS Foundation Trust, Derby, UK
²Division of Rehabilitation and Ageing, School of Medicine, University of Nottingham, Nottingham, UK
³Division of Physiotherapy and Rehabilitation Sciences, School of Health Sciences, University of Nottingham, Nottingham University Hospitals, Nottingham, UK
⁴Research Unit for General Practice in Aalborg, Department of Clinical Medicine, Aalborg University, Aalborg, Denmark
⁵Department of Occupational Therapy and Physiotherapy, Department of Clinical Medicine, Aalborg University Hospital, Aalborg, Denmark
⁶Department of Health Professions, Manchester Metropolitan University, Manchester, UK
⁷Faculty of Medicine and Health Sciences, University of East Anglia, Norwich, UK

**Twitter** @benedsmith

**Contributors** BES was responsible for conception and design, compiling the interview schedule, interviewing, transcribing, coding, analysis and interpretation, drafting and revising the manuscript. FM was responsible for conception and design, compiling the interview schedule, coding, analysis and interpretation, drafting and revising the manuscript. PH, MB, JS, MSR, TOS and PL were involved in conception and design, interpretation and reviewing revisions of the manuscript. All authors have read and approved the final manuscript.

**Funding** This report is an independent research arising from a Clinical Doctoral Research Fellowship, BES, ICA-CDRF-2015-01-002 supported by the National Institute for Health Research (NIHR) and Health Education England (HEE).

**Disclaimer** The views expressed in this publication are those of the author(s) and not necessarily those of the NHS, the NIHR, HEE or the Department of Health.

**Competing interests** None declared.

**Patient consent** Obtained.

**Ethics approval** This study was approved by the West Midlands-Black Country Research Ethics Committee (16/WM/0414).

**Provenance and peer review** Not commissioned; externally peer reviewed.

**Data sharing statement** Quotations and further details are available from Benjamin Smith at benjamin.smith3@nhs.net.

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
