## [Reviewer comments · BMJ Open]

ARTICLE DETAILS

TITLE (PROVISIONAL)	The experience of living with patellofemoral pain: loss, confusion and fear-avoidance – a UK qualitative study
AUTHORS	Smith, Benjamin; Moffatt, Fiona; Hendrick, Paul; Bateman, Marcus; Rathleff, Michael; Selfe, James; Smith, Toby O.; Logan, Phillipa

VERSION 1 – REVIEW

REVIEWER	Harald Breivik University of Oslo, Norway
REVIEW RETURNED	21-Jul-2017

GENERAL COMMENTS	This is a well planned, well done, well reported qualitative research project describing aspects of the patients' own experience with patellar pain, the patients' experience meeting health care providers, their worries and opinions
---

REVIEWER	Denis Martin Teesside University, UK
REVIEW RETURNED	03-Aug-2017

GENERAL COMMENTS	This is an interesting and worthwhile piece of work. While much work has been done from a biomechanical perspective, discussion about wider influences on and impacts of patellofemoral pain have tended to be speculative. There have been many studies that explored people's experiences of living with pain and these have generated very useful new knowledge and raised very insightful questions. This study brings that strand of research into the area of patellofemoral pain to break new ground in our understanding of this common and disabling condition. As an initial study, it does that well. Its strengths lie in raising potentially important factors that have been observed in other painful conditions that would justify a biopsychosocial approach to patellofemoral pain management. The paper has room for improvement that should be addressed so that the strengths can be seen more clearly. The reported themes lack internal coherence and often overlap to an extent that questions their individuality. For example, the first theme is called "impact on self" within which it defines a clear concept of "self". However, it then brings in separate concepts of physical activity and self-identity, and makes a strong reference to loss. To me, as a reader, that drifts from the coherence of "self" within this theme.
---

	Also, the idea of “self” reappears strongly in the theme about expectations of the future, without a clear explicit or implicit rationale why these should be presented in separation. Similarly, there are interesting references to patient-therapist interactions that appear in different themes. Also, the reporting lacks a flowing storyline that could help to link together whatever themes are chosen. The quotations do not always provide clear evidence of a theme or subtheme. For example, on page 10 the quotations offered as evidence for fear avoidance are not clearly illustrative of that phenomenon. There is no indication of fear of damage from activity and the quotes could legitimately be alternatively interpreted as a desire for rest before willingly re-engaging in activity later. There are comments related to the reflexivity of the authors that indicate their philosophy linked with the biopsychosocial model of pain and the potential for pain neurophysiology education. I personally support this philosophy but the reporting in the paper gives an impression of trying to fit the evidence into this without clear demonstration that it does. At this stage of the research, I think it would be sufficient to present evidence to plant a flag to say that this is a live possibility without veering towards trying to be conclusive and demonstrating all aspects of the phenomenon. The method is limited in two main ways. Participants were recruited using convenience sampling rather than purposive sampling, which restricts access to the full range of experiences. The issue of the limitations of sampling is noted in the paper but the line of argument in the paper is defensive about its representativeness of the population. It would be better perhaps to accept the limitation and point to how this leads to the need for further work with a more purposive sample aiming for maximum variation. In relation to sampling it may be useful to consider presenting a table outlining the main characteristics of each participant without threatening anonymity. In summary, my opinion is that this is a very good area for study with an acceptable method applied to generate novel and useful findings. I think that the analysis could be reconsidered to see if the findings could be condensed into fewer and more coherent themes presented within an intertwined narrative and using more explicitly related quotations as evidence.
--	--

REVIEWER	Dr Pirjo Vuoskoski Senior Lecturer, University of Brighton, United Kingdom
REVIEW RETURNED	02-Oct-2017

GENERAL COMMENTS	Thank you for allowing me to review this article which I found very interesting. Overall, it is written well and the premise that it would give insight into the lived experiences (or perceptions) of patellofemoral pain is very sound. However the enclosed feedback needs to be addressed before the paper would be suitable for publication. I hope my feedback will help you in improving this insightful article. Page 2/23: ABSTRACT Objectives: The aim to investigate, ‘through phenomenological inquiry’, both ‘the lived experience’ and ‘perceptions’ of ‘people with patellofemoral pain...’, perhaps, relates to the ill-defined research phenomenon and methodological approach, in the study (please see below). It is my suggestion the authors focus on exploring...
--

EITHER: 'lived experiences' (in line with the phenomenological approach), OR: 'perceptions' (still in line with a qualitative thematic analysis framework) – but the research phenomenon/interest would still need further clarification.

Design: For consistency, a critical reader would expect to see here further reference to the chosen methodological approach: 'A qualitative study design using semi-structured interview' and...

EITHER: 'phenomenological' OR: 'thematic analysis'.

Participants: In terms of methodological consistency, within a phenomenological inquiry, it would be more appropriate to use 'purposeful recruitment of participants' than 'convenience sampling methods'.

Results: Related to earlier comments; within a phenomenological framework, the meaning and (meaningfully related) implications (rather than the impact) of pain would be typically addressed, in terms of the nature of the (qualitative) data and interpretation of the results.

ARTICLE SUMMARY

Strengths and limitations of the study:

Line 53: The claim for being the 'first study to use a qualitative method of inquiry', in the present form of the article, may be acceptable, but not for 'gaining phenomenological data'.

Lines 55-56: This claim again supports a suggestion to amend the methodological framework to a 'qualitative thematic (content) analysis'.

Page 3/23:

Lines 3-6: This paragraph again is not congruent with a phenomenological framework; when adopting a phenomenological attitude and method (interpretive or descriptive), the aim is not to 'generalise from a sample to a population'; therefore, it would not be appropriate to evaluate the study with the same (quantitative research) criteria either (for example, to evaluate how well the sample 'represents' the population'). However, some qualitative (e.g. 'content analysis') approaches may copy the mainstream paradigm approach, including the use of criteria and terminology; this again suggests the need for amending the methodological framework.

Page 4/23:

INTRODUCTION:

Although the section works well in establishing the context of the work being reported, in terms of discussing the relevant literature and summarising current understanding of the area of interest, it is lacking clarification of the research phenomenon/interest (as understood in phenomenology) and rationale for the chosen (phenomenological) approach. This again relates to the problems highlighted above, and suggests the need for either amending the methodological framework or adding further clarification of the above-mentioned aspects.

Page 5/23:

METHOD:

This section has similar problems (as addressed above), and again suggests the need to amend the methodological framework - perhaps to a 'qualitative thematic (content) analysis'; especially due to the current choice of the method of analysis (Braun and Clarke, thematic analysis), and its (lack of) justification. In the current manuscript, no claim is being made that a phenomenological attitude, reflection or method, has been implemented in the study (other than a short reference to the use of reflective journal in the data collection and analysis sections, which also require further clarification);

and whether interpretive or descriptive line of phenomenological tradition, for example, were followed. In this section, a critical reader would anticipate some explanation of whether/how the researcher implemented phenomenological reflection, based on 'reflexivity' or 'bracketing', in line with the chosen approach; and why 'thematic analysis' as presented by Braun and Clarke was considered as 'the most appropriate method for this type of inquiry'. Therefore, further clarification is required.

Although, it could be claimed that the current study would be better suited under a more 'generic' qualitative research design (perhaps, interpretivist, which again would need more clarification), building on semi-structured interviews and thematic analysis (which would be thus more appropriate design for the manuscript, in its current form), further clarification of the adopted methodological stance is still required; for example, what is meant by 'the contextualist position' (and 'sitting in the central on the spectrum of realism and constructivism') in the study; which again should be in line with the understanding of the research phenomenon/interest and rationale (see the comments in the 'introduction' section).

PARTICIPANTS / RECRUITMENT

Overall, it seems that the section 'Participants' largely consists of information that would be better suited under the following section (Recruitment). And the issues under 'Recruitment' are mainly related to 'Data collection'. Hence, it is my suggestion that the sections ('participants' and 'recruitment') would be combined into one, and that most of the text under 'recruitment' would be removed to 'Data collection'.

Please see the earlier comments related to participant recruitment/sampling methods.

Line 25: 'Anticipations related to sample size and data saturation', are not issues typically addressed in a phenomenologically grounded study. Therefore, if claiming a phenomenological status, further clarification/justification is required.

It is not clearly stated in this section, who initially contacted the participants, who did the telephone follow-up, whether/how coercion was considered when recruiting the participants, and when/how the exclusion criteria were reviewed/screened. Further clarification is required.

DATA COLLECTION:

For the critical reader (particularly within the phenomenological tradition), to be able to evaluate the quality of the interview, the interview plan and how the 'topic guidelines' mentioned by the authors were implemented in practice would need further clarification. For example, the current text now creates an impression that there may have been a lack of spontaneity and openness in the interview as expected in a phenomenological interview; for example, if the topics of participant's 'interpretation of the causation of their pain' or 'specific coping strategies' were applied as interview questions; this may have led the interviewee to say certain specific things that the researchers were seeking in the data, and thus could be considered an example of biasing the data. It is therefore suggested that further clarification would be added to the text/appendices. In addition, the reader would expect further clarification of the aim/use of the reflective journal (specifically if the phenomenological attitude/approach will be maintained; please see the comments in the 'Method' section). In addition, how the interviews were (audio?)recorded needs a brief clarification.

Page 6/23:

DATA ANALYSIS:

Overall, this section (in terms of being a qualitative, thematic analysis) is well written. However, how exactly (when and by whom) the data were transcribed, and the aim and use of the reflective journal, need further clarification (for example, linked to reflection of the researcher's position, bias etc.). Furthermore, the aim and use of NVivo (for data management/analysis?) should be explained more in detail, and a better justification and rationale for implementing the ideas of data 'saturation', and 'generation' of 'new thoughts or concepts', is required. In addition, the process of how the 'themes' were developed should be made more transparent for a critical reader. For example, it could be presented using both text and illustrative materials (e.g. table/s and/or figures).

Page 7-13/23:

RESULTS:

Firstly, more information (other than mean and range) about the research context, the participants and their background, would be beneficial (for example, their profession, if they were working full-time/part-time, or on sick leave, if they had a family/partner or children, hobbies etc.), if possible. This could be presented using both text and illustrative materials (e.g. table/s).

Secondly, although this section, with a lot of reference to the raw data, is rather illuminating in terms of the main themes as being present in the data, it is quite challenging to read, and there are a few issues that would require further consideration/amending. In general, the section could be written more concisely, with a better synthesis of the key findings, perhaps organised around tables or figures (e.g. presenting the meaningful relations of the key themes). Furthermore, there could be more signposting and linking between paragraphs to guide the reader through the results.

Thirdly, the results (in this section) are discussed in light of the data as well as existing literature. My suggestion would be to focus on the key findings of this study (as presented in the data) first, with reference to raw data (as it is now), and to remove all discussion with the literature to the following section (Discussion). This would add more clarity to the key findings and the knowledge contribution in this study.

Lastly, while going through the current manuscript, the critical reader cannot avoid the impression that the data analysis may have been building on theoretical 'lenses' (e.g. interpreting the results based on previous knowledge and pre-assumptions);

For example:

Page 10, Theme 3, lines 37-40 ('Associated with cultural beliefs on pain and damage was the resultant fear-avoidant behaviour...')

Page 12, Theme 5, lines 13-14 (participants expressing an 'external locus of control'), lines 42-43 (participants holding 'negative beliefs about the future' and 'prognosis' being 'low self-efficacy').

Therefore, the considerations/reflections on the researcher's position, pre-assumptions and bias, in the study becomes even more important (please see the earlier comments).

Pages 13-15/23:

DISCUSSION:

Overall, this section could be improved in terms of interpretation of the results, what was already known, and addressing the significance of the key findings. In some parts, for example, the discussion in terms of previous literature (e.g. the potential link between the patient exposure to so-called biomedical models, diagnostic or causal explanations of PFP, and their lived experience) does not seem to address the key meanings (as being present in the patient experience), in terms of a phenomenological approach.

	This again, is in line with the previous comments about the methodological decision making in this study. More detailed comment: Lines 35-36: This claim (of obtaining phenomenological data) and its justification raise expectations of the choice, rationale and rigor of the methodological approach in this study, as already has been raised in the previous comments. Page 15/23: CLINICAL AND RESEARCH IMPLICATIONS / STUDY LIMITATIONS AND STRENGTHS: To avoid unnecessary repetition, please see the previous comments related to the use of the mainstream research criteria and terminology (e.g. to the article summary, introduction, and methods sections), and addressing the significance of the significance of the findings (and their implications) in this study. More detailed comment: Lines 31-32: This claim (providing a clear, transparent, and reproducible methodological approach to data analysis) again raises expectations of providing a transparent illustration of the research analysis, as raised in the previous comments (to data analysis section). Pages 15-16: CONCLUSIONS: Please see the previous comment about addressing the meaning of the results in this study more clearly. Pages 17-20: REFERENCES: Although the reference list first seems appropriate and up-to-date, I would expect at least a few references more directly related to the phenomenological approach and methodological underpinnings, which would be then referenced in the main text. This again obviously relates to the final methodological decision making.
--	---

VERSION 1 – AUTHOR RESPONSE

Reviewer: 1

Reviewer Name: Harald Breivik

Institution and Country: University of Oslo, Norway Competing Interests: None

This is a well planned, well done, well reported qualitative research project describing aspects of the patients' own experience with patellar pain, the patients' experience meeting health care providers, their worries and opinions

Response: Thank you for your kind comments.

Reviewer: 2

Reviewer Name: Denis Martin

Institution and Country: Teesside University, UK Competing Interests: None declared

Comment: This is an interesting and worthwhile piece of work. While much work has been done from a biomechanical perspective, discussion about wider influences on and impacts of patellofemoral pain have tended to be speculative. There have been many studies that explored people's experiences of living with pain and these have generated very useful new knowledge and raised very insightful questions. This study brings that strand of research into the area of patellofemoral pain to break new ground in our understanding of this common and disabling condition. As an initial study, it does that well. Its strengths lie in raising potentially important factors that have been observed in other painful conditions that would justify a biopsychosocial approach to patellofemoral pain management.

The paper has room for improvement that should be addressed so that the strengths can be seen more clearly.

The reported themes lack internal coherence and often overlap to an extent that questions their individuality. For example, the first theme is called “impact on self” within which it defines a clear concept of “self”. However, it then brings in separate concepts of physical activity and self-identity, and makes a strong reference to loss. To me, as a reader, that drifts from the coherence of “self” within this theme. Also, the idea of “self” reappears strongly in the theme about expectations of the future, without a clear explicit or implicit rationale why these should be presented in separation. Similarly, there are interesting references to patient-therapist interactions that appear in different themes. Also, the reporting lacks a flowing storyline that could help to link together whatever themes are chosen.

Response: On reflection, and on taking into consideration positive comments from reviewers 1 and 3, we have decided to keep the themes in their current format. The themes represent our interpretation of the results, and we acknowledge overlap of some elements of the themes, particularly in relation to self. We have added further signposting at the start of the results section to improve the narrative, and help the reader understand our rationale.

“The first theme that emerged from the data, impact on self, describes the participants’ sense of loss, in relation to their self and self-identity. The further themes that emerged describe how the participants deal with this loss in a climate of uncertainty, how they understand or make decisions regarding exercise/activity and pain management, and how they prognosticate for the future. Data are presented to demonstrate the range and meaning to each theme.”

We have also tidied and cleaned the first theme to improve the internal coherency.

Comment: The quotations do not always provide clear evidence of a theme or subtheme. For example, on page 10 the quotations offered as evidence for fear avoidance are not clearly illustrative of that phenomenon. There is no indication of fear of damage from activity and the quotes could legitimately be alternatively interpreted as a desire for rest before willingly re-engaging in activity later.

Response: We have re-worded this sentence to improve clarity:

“Participants, frequently contradicted themselves however; many participants would express the sentiment that they would not let the pain stop them from doing what they wanted to do, yet demonstrated clear activity withdrawal.

‘So for example, we went to [holiday resort] last year; on your feet all day, walking miles and miles, I would be, like, in tears by the end of the day. I wouldn’t let it stop me the next day because I would be, like, I’m doing this’ [P4].

‘When I was in [holiday resort]; a couple of days I didn’t go out and I stayed back at the hotel. Because I couldn’t do it, I needed to rest.’ [P4].”

Comment: There are comments related to the reflexivity of the authors that indicate their philosophy linked with the biopsychosocial model of pain and the potential for pain neurophysiology education. I personally support this philosophy but the reporting in the paper gives an impression of trying to fit the evidence into this without clear demonstration that it does. At this stage of the research, I think it would be sufficient to present evidence to plant a flag to say that this is a live possibility without veering towards trying to be conclusive and demonstrating all aspects of the phenomenon.

Response: We have reflected on your comments, and have removed the sentences with regards to biopsychosocial treatments and pain neurophysiology education in the clinical implications section of the discussion.

Comment: The method is limited in two main ways. Participants were recruited using convenience sampling rather than purposive sampling, which restricts access to the full range of experiences. The issue of the limitations of sampling is noted in the paper but the line of argument in the paper is defensive about its representativeness of the population. It would be better perhaps to accept the limitation and point to how this leads to the need for further work with a more purposive sample aiming for maximum variation. In relation to sampling it may be useful to consider presenting a table outlining the main characteristics of each participant without threatening anonymity.

Response: As suggest, we have removed the sentence on the studies representativeness of the population from the limitation section of the discussion.

We have also added a table outlining the main characteristics of each participant.

Comment: In summary, my opinion is that this is a very good area for study with an acceptable method applied to generate novel and useful findings. I think that the analysis could be reconsidered to see if the findings could be condensed into fewer and more coherent themes presented within an intertwined narrative and using more explicitly related quotations as evidence.

Response: We have added further signposting at the start of the results section to improve the intertwined narrative, and have tidied up and changed one of the quotations in relation to fear avoidance.

Reviewer: 3

Reviewer Name: Dr Pirjo Vuoskoski

Institution and Country: Senior Lecturer, University of Brighton, United Kingdom Competing Interests: None declared

Comment: Thank you for allowing me to review this article which I found very interesting. Overall, it is written well and the premise that it would give insight into the lived experiences (or perceptions) of patellofemoral pain is very sound. However the enclosed feedback needs to be addressed before the paper would be suitable for publication. I hope my feedback will help you in improving this insightful article. Please see the attached file.

Page 2/23:

ABSTRACT

Objectives: The aim to investigate, 'through phenomenological inquiry', both 'the lived experience' and 'perceptions' of 'people with patellofemoral pain...', perhaps, relates to the ill-defined research phenomenon and methodological approach, in the study (please see below). It is my suggestion the authors focus on exploring... EITHER: 'lived experiences' (in line with the phenomenological approach), OR: 'perceptions' (still in line with a qualitative thematic analysis framework) – but the research phenomenon/interest would still need further clarification.

Response: Thank you for taking the time to read and comment on our paper. We have taken your comments on board and have revised our methodological approach to better reflect the study. We have re written the approach that now describes an interpretivist perspective where we explore people's perceptions, experiences and beliefs. This has been reflected in a new title, with changes to the abstract, introduction and methods sections, respectively.

Design: For consistency, a critical reader would expect to see here further reference to the chosen methodological approach: 'A qualitative study design using semi-structured interview' and... EITHER: 'phenomenological' OR: 'thematic analysis'.

Response: We have now added the following sentence:

“Qualitative study design using semi-structured interviews, and analysed thematically using the guidelines set out by Braun and Clarke.”

Participants: In terms of methodological consistency, within a phenomenological inquiry, it would be more appropriate to use 'purposeful recruitment of participants' than 'convenience sampling methods'.

Response: This comment has been addressed by our revision of our methodological approach.

Results: Related to earlier comments; within a phenomenological framework, the meaning and (meaningfully related) implications (rather than the impact) of pain would be typically addressed, in terms of the nature of the (qualitative) data and interpretation of the results.

Response: This comment has been addressed by our revision of our methodological approach.

ARTICLE SUMMARY

Strengths and limitations of the study:

Line 53: The claim for being the 'first study to use a qualitative method of inquiry', in the present form of the article, may be acceptable, but not for 'gaining phenomenological data'.

Response: This has been corrected to: "This is the first study to use a qualitative method of inquiry on experience of people living with patellofemoral pain."

Lines 55-56: This claim again supports a suggestion to amend the methodological framework to a 'qualitative thematic (content) analysis'.

Response: This comment has been addressed by our revision of our methodological approach.

Page 3/23:

Lines 3-6: This paragraph again is not congruent with a phenomenological framework; when adopting a phenomenological attitude and method (interpretive or descriptive), the aim is not to 'generalise from a sample to a population'; therefore, it would not be appropriate to evaluate the study with the same (quantitative research) criteria either (for example, to evaluate how well the sample 'represents' the population'). However, some qualitative (e.g. 'content analysis') approaches may copy the mainstream paradigm approach, including the use of criteria and terminology; this again suggests the need for amending the methodological framework.

Response: This has been corrected to: "For pragmatic reasons a convenience sampling technique was used."

Page 4/23:

INTRODUCTION:

Although the section works well in establishing the context of the work being reported, in terms of discussing the relevant literature and summarising current understanding of the area of interest, it is lacking clarification of the research phenomenon/interest (as understood in phenomenology) and rationale for the chosen (phenomenological) approach. This again relates to the problems highlighted above, and suggests the need for either amending the methodological framework or adding further clarification of the above-mentioned aspects.

Response: This comment has been addressed by our revision of our methodological approach.

Page 5/23:

METHOD:

This section has similar problems (as addressed above), and again suggests the need to amend the methodological framework - perhaps to a 'qualitative thematic (content) analysis'; especially due to the current choice of the method of analysis (Braun and Clarke, thematic analysis), and its (lack of) justification. In the current manuscript, no claim is being made that a phenomenological attitude, reflection or method, has been implemented in the study (other than a short reference to the use of reflective journal in the data collection and analysis sections, which also require further clarification); and whether interpretive or descriptive line of phenomenological tradition, for example, were followed.

In this section, a critical reader would anticipate some explanation of whether/how the researcher implemented phenomenological reflection, based on 'reflexivity' or 'bracketing', in line with the chosen approach; and why 'thematic analysis' as presented by Braun and Clarke was considered as 'the most appropriate method for this type of inquiry'. Therefore, further clarification is required.

Although, it could be claimed that the current study would be better suited under a more 'generic' qualitative research design (perhaps, interpretivist, which again would need more clarification), building on semi-structured interviews and thematic analysis (which would be thus more appropriate design for the manuscript, in its current form), further clarification of the adopted methodological stance is still required; for example, what is meant by 'the contextualist position' (and 'sitting in the central on the spectrum of realism and constructivism') in the study; which again should be in line with the understanding of the research phenomenon/interest and rationale (see the comments in the 'introduction' section).

Response: We have now revised the methods sections, taking into account the revised methodological approach.

"In order to address gaps in the literature this research focused on identifying themes within the participants' experience of living with PFP. A qualitative interpretive description design was chosen as an appropriate methodological approach.^[26] Thematic analysis is the most appropriate method for this type of inquiry, as codes and themes can be created inductively to capture meaning and content without prior preconceptions allowing flexibility to generate a rich and detailed account of the data.^[27] "

In this study, data were analysed thematically using the guidelines set out by Braun and Clarke,^[27] and was reported in line with the COnsolidated criteria for REporting Qualitative research (COREQ) checklist (see supplementary file 1).^[28]

Braun and Clarke^[27] describe a multi-stage approach to thematic data analysis; demonstrating clear distinction of the thematic approach, whilst allowing for the inherent flexibility in the process. They reasoned that a thematic analysis can be conducted from a both realist and constructionist paradigms, although with differing outcomes. A realist approach allows theories about individual motivation and meaning to be developed, since the epistemological position is that there is a unidirectional relationship between meaning, experience and language^[27]. A constructionist perspective differs, as meaning and experience are socially produced and knowledge a human and social construct; therefore theories about individual motivation and meaning are inappropriate, and theories focus instead on sociocultural contexts^[27]. This study did not set out to prove or disprove as hypotheses; it set out to generate new data from which an understanding of living with PFP might be developed. The authors wanted to take an epistemological position that recognises the experience at an individual level, and any meanings attached, whilst considering the wider context within a sociocultural perspective. Sitting central on the spectrum of realism and constructivism, this position is described as "contextualist" by Braun and Clarke^[27]."

PARTICIPANTS / RECRUITMENT

Overall, it seems that the section 'Participants' largely consists of information that would be better suited under the following section (Recruitment). And the issues under 'Recruitment' are mainly related to 'Data collection'. Hence, it is my suggestion that the sections ('participants' and 'recruitment') would be combined into one, and that most of the text under 'recruitment' would be removed to 'Data collection'.

Response: We have revised the manuscript as suggested.

Comment: Please see the earlier comments related to participant recruitment/sampling methods.

Response: This comment has been addressed by our revision of our methodological approach.

Line 25: 'Anticipations related to sample size and data saturation', are not issues typically addressed in a phenomenologically grounded study. Therefore, if claiming a phenomenological status, further clarification/justification is required.

Response: This comment has been addressed by our revision of our methodological approach.

Comment: It is not clearly stated in this section, who initially contacted the participants, who did the telephone followup, whether/how coercion was considered when recruiting the participants, and when/how the exclusion criteria were reviewed/screened. Further clarification is required.

Response: We have now added author initials for the recruitment procedure.

Our manuscript stated that the inclusion criteria was pre-screened during an initial telephone conversation. We have now added details for the exclusion criteria. "The exclusion criteria were screened prior to consent being taken."

DATA COLLECTION:

For the critical reader (particularly within the phenomenological tradition), to be able to evaluate the quality of the interview, the interview plan and how the 'topic guidelines' mentioned by the authors were implemented in practice would need further clarification. For example, the current text now creates an impression that there may have been a lack of spontaneity and openness in the interview as expected in a phenomenological interview; for example, if the topics of participant's 'interpretation of the causation of their pain' or 'specific coping strategies' were applied as interview questions; this may have led the interviewee to say certain specific things that the researchers were seeking in the data, and thus could be considered an example of biasing the data. It is therefore suggested that further clarification would be added to the text/appendices. In addition, the reader would expect further clarification of the aim/use of the reflective journal (specifically if the phenomenological attitude/approach will be maintained; please see the comments in the 'Method' section). In addition, how the interviews were (audio?) recorded needs a brief clarification.

Response: This comment has been part addressed by our revision of our methodological approach. In addition we have added some further clarification, with the addition of the following sentences:

"The semi-structured interviews allowed for a flexible interview, in a two-way conversation, allowing new ideas to be developed as they were brought up. "

Page 6/23:

DATA ANALYSIS:

Overall, this section (in terms of being a qualitative, thematic analysis) is well written. However, how exactly (when and by whom) the data were transcribed, and the aim and use of the reflective journal, need further clarification (for example, linked to reflection of the researcher's position, bias etc.).

Furthermore, the aim and use of NVivo (for data management/analysis?) should be explained more in detail, and a better justification and rationale for implementing the ideas of data 'saturation', and 'generation' of 'new thoughts or concepts', is required. In addition, the process of how the 'themes' were developed should be made more transparent for a critical reader. For example, it could be presented using both text and illustrative materials (e.g. table/s and/or figures).

Response: This comment has been part addressed by our revision of our methodological approach. In addition we have added initials for the transcribing.

Page 7-13/23:

RESULTS:

Firstly, more information (other than mean and range) about the research context, the participants and their background, would be beneficial (for example, their profession, if they were working full-time/parttime, or on sick leave, if they had a family/partner or children, hobbies etc.), if possible. This could be presented using both text and illustrative materials (e.g. table/s).

Response: We have now added a table outlining the main characteristics of each participant. Unfortunately some of the background you asked for is not within our data set. We would also be concerned with some of the data you requested with regards to participant anonymity.

Comment: Secondly, although this section, with a lot of reference to the raw data, is rather illuminating in terms of the main themes as being present in the data, it is quite challenging to read, and there are a few issues that would require further consideration/amending. In general, the section could be written more concisely, with a better synthesis of the key findings, perhaps organised around tables or figures (e.g. presenting the meaningful relations of the key themes). Furthermore, there could be more signposting and linking between paragraphs to guide the reader through the results.

Response: We have tidied and cleaned the first theme to improve the internal coherency, with further signposting leading into this.

With reference to previous published work in the BMJ Open, it is common practice to provide some discussion for each theme, within the results. With the main discussion of the paper drawing on main discussion point. Considering the feedback from the other reviewers we are committed to keeping the results predominantly in their current form. We feel this provides sufficient narrative for each theme.

Comment: Thirdly, the results (in this section) are discussed in light of the data as well as existing literature. My suggestion would be to focus on the key findings of this study (as presented in the data) first, with reference to raw data (as it is now), and to remove all discussion with the literature to the following section (Discussion). This would add more clarity to the key findings and the knowledge contribution in this study.

Response: Please see above answer.

Comment: Lastly, while going through the current manuscript, the critical reader cannot avoid the impression that the data analysis may have been building on theoretical 'lenses' (e.g. interpreting the results based on previous knowledge and pre-assumptions);

For example:

Page 10, Theme 3, lines 37-40 ('Associated with cultural beliefs on pain and damage was the resultant fearavoidant behaviour...')

Page 12, Theme 5, lines 13-14 (participants expressing an 'external locus of control'), lines 42-43 (participants holding 'negative beliefs about the future' and 'prognosis' being 'low self-efficacy').

Therefore, the considerations/reflections on the researcher's position, pre-assumptions and bias, in the study becomes even more important (please see the earlier comments).

Response: We addressed our potential research bias, and discuss this in the strengths and limitations of the paper.

"The authors make it clear that their clinical and research experience lie within the biopsychosocial framework of musculoskeletal pain and this study forms part of a larger body of research looking at pain education, self-management strategies and exercise interventions for individuals with PFP."

Pages 13-15/23:

DISCUSSION:

Overall, this section could be improved in terms of interpretation of the results, what was already known, and addressing the significance of the key findings. In some parts, for example, the discussion in terms of previous literature (e.g. the potential link between the patient exposure to so-called biomedical models, diagnostic or causal explanations of PFP, and their lived experience) does not seem to address the key meanings (as being present in the patient experience), in terms of a phenomenological approach. This again, is in line with the previous comments about the methodological decision making in this study.

Response: This comment has been addressed by our revision of our methodological approach.

More detailed comment:

Lines 35-36: This claim (of obtaining phenomenological data) and its justification raise expectations of the choice, rationale and rigor of the methodological approach in this study, as already has been raised in the previous comments.

This comment has been addressed by our revision of our methodological approach.

Page 15/23:

CLINICAL AND RESEARCH IMPLICATIONS / STUDY LIMITATIONS AND STRENGTHS:

To avoid unnecessary repetition, please see the previous comments related to the use of the mainstream research criteria and terminology (e.g. to the article summary, introduction, and methods sections), and addressing the significance of the findings (and their implications) in this study.

Response: This comment has been addressed by our revision of our methodological approach.

More detailed comment:

Lines 31-32: This claim (providing a clear, transparent, and reproducible methodological approach to data analysis) again raises expectations of providing a transparent illustration of the research analysis, as raised in the previous comments (to data analysis section).

This comment has been addressed by our revision of our methodological approach.

Pages 15-16:

CONCLUSIONS:

Please see the previous comment about addressing the meaning of the results in this study more clearly.

Response: This comment has been addressed by our revision of our methodological approach.

Pages 17-20:

REFERENCES:

Although the reference list first seems appropriate and up-to-date, I would expect at least a few references more directly related to the phenomenological approach and methodological underpinnings, which would be then referenced in the main text. This again obviously relates to the final methodological decision making.

Response: This comment has been addressed by our revision of our methodological approach.

VERSION 2 – REVIEW

REVIEWER	Denis Martin Teesside University, UK.
REVIEW RETURNED	28-Oct-2017

GENERAL COMMENTS	The authors have revised the paper well.
--

REVIEWER	Dr Pirjo Vuoskoski University of Brighton, UK
REVIEW RETURNED	10-Nov-2017

GENERAL COMMENTS	Thank you for allowing me to review the much improved manuscript. However, in my view, a couple of issues would still require further clarification/amending: Recruitment: 1. As already addressed in the first round, further clarification is required on how the researcher considered potential coercion of participants: Ideally the recruitment email would come from someone else than the researcher, and ask the person to call/send an email for additional information, if interested in participating in the study. In addition, participants were recruited from the same thrust (even same clinic?), where the researcher/interviewer works as a physiotherapist. These (ethical issues) should at least be acknowledged as limitations of the study. 2. Further clarification is required, what exactly resulted to recruitment of 10 participants; and whether data 'saturation' was linked to it (as you imply later in the data analysis section). Data analysis: Further clarification is required on issues also addressed in the first round: 1. How exactly NVivo was used in the study (for data management and/or analysis?) needs further clarification.2. The analysis process requires further clarification/more transparency; demonstration of how exactly the themes were developed. Although the steps of the process have been explained, the critical other needs to be able to follow the audit trail of the process; formation of the themes and sub-themes (clearly presented, e.g. with the help of illustrative materials; table/s and/or figures).
---

	Study limitations and strengths: This section requires further amending, as there may be more limitations to consider. See the previous comments on:  1. There is no clear audit trail of the analysis in this study for the critical other – which is a necessary requirement for 'transparency', in a qualitative data analysis. 2. The participant recruitment process (and its link to data saturation) is still not clearly explained in the study. Conclusion: Use the terminology consistently throughout (with the title, aim of the research etc.). Please see this comments in the enclosed manuscript.
--	---

VERSION 2 – AUTHOR RESPONSE

Reviewers' Comments to Author:

Reviewer: 2

Reviewer Name: Denis Martin

Institution and Country: Teesside University, UK.

Competing Interests: None.

The authors have revised the paper well.

No response needed.

Reviewer: 3

Reviewer Name: Dr Pirjo Vuoskoski

Institution and Country: University of Brighton, UK Competing Interests: None declared

Thank you for allowing me to review the much improved manuscript.

However, in my view, a couple of issues would still require further clarification/amending:

Recruitment:

1. As already addressed in the first round, further clarification is required on how the researcher considered potential coercion of participants:

Ideally the recruitment email would come from someone else than the researcher, and ask the person to call/send an email for additional information, if interested in participating in the study.

In addition, participants were recruited from the same thrust (even same clinic?), where the researcher/interviewer works as a physiotherapist.

These (ethical issues) should at least be acknowledged as limitations of the study.

Response: We have adjusted the first paragraph in the strengths and limitations sections to better convey this potential limitation:

“It is worth noting that the interviewer made it explicit to the participants that he was a physiotherapist working in the department conducting the research; indeed a number of them did proceed to ask clinical questions about their condition, highlighting a power dynamic between the interviewer and participant. Furthermore, it is important to note that recruitment took place in the same department that the researcher was working as a physiotherapist. This may, in part, have influenced their participants’ inclination to take part, and also their responses.”

2. Further clarification is required, what exactly resulted to recruitment of 10 participants; and whether data ‘saturation’ was linked to it (as you imply later in the data analysis section).

Response: We have adjusted the recruitment section’s first sentence to read:

“A convenience sample of ten participants with a diagnosis of PFP were recruited from an NHS physiotherapy waiting list. Based on similar studies of other musculoskeletal conditions, we anticipated this sample size would be sufficient to reach data saturation, and was agreed a priori”

Data analysis:

Further clarification is required on issues also addressed in the first round:

1. How exactly NVivo was used in the study (for data management and/or analysis?) needs further clarification.

The sentence currently reads “Data were organised and analysed using QSR International's NVivo 11.”

Response: We believe this already answers your query.

2. The analysis process requires further clarification/more transparency; demonstration of how exactly the themes were developed.

Although the steps of the process have been explained, the critical other needs to be able to follow the audit trail of the process; formation of the themes and sub-themes (clearly presented, e.g. with the help of illustrative materials; table/s and/or figures).

Response: We feel our clear and detailed description of our analysis methods in the methods sections is actually a strength of our paper, demonstrating a clear audit trail of the process. However, we have now added a supplementary file of our code book, produced by NVivo, which we trust satisfies the reviewer.

Study limitations and strengths:

This section requires further amending, as there may be more limitations to consider.

See the previous comments on:

1. There is no clear audit trail of the analysis in this study for the critical other – which is a necessary requirement for 'transparency', in a qualitative data analysis.

2. The participant recruitment process (and its link to data saturation) is still not clearly explained in the study.

Response: These two points have now been addressed.

Conclusion:

Use the terminology consistently throughout (with the title, aim of the research etc.).

Response: We have now revised the first sentence, so it is now consistent with the title. "These findings offer an insight into the experience of individuals living with PFP."

VERSION 3 – REVIEW

REVIEWER	Pirjo Vuoskoski University of Brighton, UK
REVIEW RETURNED	17-Nov-2017
GENERAL COMMENTS	I thank the authors for their respond to the previous comments, and providing a revised manuscript. However, I still think that the manuscript could be strengthened and would benefit from further revision. I am happy with the 'strengths and limitations' section, conveying the ethical issues in participant recruitment, which cannot be changed afterwards. On the other hand, it would have been better to remove all reference to 'data saturation', since no further evidence of this is presented. This relates to my expectations for further transparency; (although explained well in theory) how exactly (and how rigorously and systematically) the methods were implemented in this study; this refers to an audit trail (the steps of the analysis illuminated with reference to the data) of the analysis process, which still is not provided by the authors. Presenting a list of codes does not illuminate the iterative process of how exactly the themes and sub-themes were developed and generated from the raw data. Therefore, my opinion is that this study could be further improved and strengthened. However, I am inclined to leave the decision for the editor.

VERSION 3 – AUTHOR RESPONSE

We thank the editor and reviewers for taking their time to carefully read our manuscript again.

We have responded to the comments below point by point (text in red).

I thank the authors for their respond to the previous comments, and providing a revised manuscript.

However, I still think that the manuscript could be strengthened and would benefit from further revision.

I am happy with the 'strengths and limitations' section, conveying the ethical issues in participant recruitment, which cannot be changed afterwards.

On the other hand, it would have been better to remove all reference to 'data saturation', since no further evidence of this is presented.

Response: This is not correct. We reference 'data saturation' in two sections; in the recruitment section and data analysis section. Data saturation is an important point, since based on similar studies of other musculoskeletal conditions, we anticipated this sample size would be sufficient to reach data saturation, and was agreed a priori. Also, 2 out of 3 reviewers were satisfied with our reference to 'data saturation', therefore we have kept this in the manuscript in its current form.

This relates to my expectations for further transparency; (although explained well in theory) how exactly (and how rigorously and systematically) the methods were implemented in this study; this refers to an audit trail (the steps of the analysis illuminated with reference to the data) of the analysis process, which still is not provided by the authors. Presenting a list of codes does not illuminate the iterative process of how exactly the themes and sub-themes were developed and generated from the raw data.

Therefore, my opinion is that this study could be further improved and strengthened. However, I am inclined to leave the decision for the editor.

Response: The methods sections described the steps, not just in theory, but our actual method of undertaking the analysis. We have no further data to provide or add, and the manuscript has remained in its current format for this revision.